# Evaluation of the influence of correcting for gillnet selectivity on the estimation of population parameters

**Zachary Klein** [1]◉*, **Joshua McCormick** [2]◉

**1** Department of Fish, Wildlife, and Conservation Ecology, New Mexico State University, Las Cruces, New Mexico, United States of America, **2** Idaho Department of Fish and Game, Nampa, Idaho, United States of America

◉ These authors contributed equally to this work.
* Zklein@nmsu.edu

## Abstract

Gill nets are a common sampling technique in inland and marine fisheries. However, gill nets are size selective and may result in bias estimates of population parameters. As such, selectivity is commonly assessed using indirect estimation techniques. Indirect estimates of gillnet selectivity have been suggested to improve estimates of important populations metrics (e.g., total annual mortality), but this assertion has not been assessed. In the current study, we simulated hypothetical populations of channel catfish *Ictalurus punctatus*, lake trout *Salvelinus namaycush*, walleye *Sander vitreus*, and white crappie *Pomoxis annularis* and sampled the populations based on published gillnet encounter and retention probabilities. Total annual mortality and von Bertalanffy parameters were then estimated using unadjusted (not "correcting" for selectivity processes) and adjusted ("correcting" for selectivity processes) age and(or) length data to evaluate the value of accounting for gillnet selectivity when estimating these metrics. Our results indicate that adjusting for retention and encounter probabilities rarely leads to improved estimates of total annual mortality, $K$, and $L_\infty$. For instance, estimates of annual mortality of lake trout based on age data adjusted for retention probability resulted in an overestimate of $A$ by 14.4%. As such, we suggest that analysis of gillnet selectivity only be used when specific questions are being addressed (e.g., catch-at-age models) or in situation when all processes contributing to gillnet selectivity (e.g., contact probability, size-specific availability) are known.

## Introduction

Gill nets are a common sampling technique for surveying and assessing fish populations [1]. An underlying assumption of gill nets (and most sampling gear) is that the sample data are representative of the target populations for which inference is made (i.e., proportionality [2, 3]). However, the assumption of proportionality is rarely achieved due to the size selectivity of gill nets [1, 3]. Violations of this assumption may have important ramifications for commonly used population metrics (e.g., catch per unit effort, annual mortality) and can lead to

**Data Availability Statement:** The analysis in the manuscript is a simulation of four hypothetical fish populations. These populations were simulated using data from Tables 1 and 2 in the manuscript. Therefore, all the data needed to recapitulate the

analysis can be found in Tables 1 and 2. No additional data are associated with the analysis in the current paper.

**Funding:** The authors received no specific funding for this work.

**Competing interests:** No authors have competing interests

erroneous conclusions about fish populations [4]. As such, numerous authors have suggested accounting for size selectivity of gill nets to accurately represent the true population structure of a target fish population [3, 5–7].

Gillnet size selectivity can be estimated using direct or indirect methods. Direct selectivity methods estimate capture probability based on the known size distribution of a population (or subpopulation [5, 8, 9]). However, direct estimation is often challenging due to the difficulty of censusing a population and mark-recapture [5, 10]. For instance, mark-recapture techniques are often difficult or impossible with gears that exhibit high associated mortality (e.g., mid-water trawls, gill nets [10]). Therefore, gillnet selectivity is most often estimated using indirect methods.

Indirect methods for estimating gillnet selectivity do not require knowledge of the true population structure. Instead, selectivity is most often modeled as the relative probability that a fish of length *l* will be retained in a gill net given that it contacts the gill net (retention probability [5, 11, 12]). Retention selectivity curves are developed by simultaneously fishing multiple mesh sizes (e.g., experimental gill net) and calculating the proportion of the total catch of a length class across all available mesh sizes [5, 6]. The mesh size with the highest catch rate is then used to scale the relative probability of capture of all other mesh sizes. Estimation of relative retention probability itself is generally not the objective of the gillnet selectivity estimation process. Rather, estimation of relative retention probability is an intermediate step that allows for estimation of the "true" length (or age) distribution of a fish population [7, 13–15]. Although it is commonly assumed that correcting for gillnet selectivity leads to more accurate length structure data, this assumption is rarely (if ever) assessed.

Although relative retention probabilities have been widely used to approximate gillnet selectivity, retention probability alone does not encompass all of the processes responsible for gillnet selectivity [5, 6, 12]. For instance, Anderson [16] delineated the capture process in gill nets as the probability of encountering a net, the probability of contacting a net, and the probability of being retained in a net. Encounter probability, in particular, likely plays an important role in the selectivity of gill nets [17–19]. Estimating encounter probabilities is challenging because it incorporates difficult-to-quantify fish behaviors that can be indirectly influenced by factors such as season and habitat [12, 17, 20]. Notwithstanding, some authors have attempted to incorporate encounter probabilities in gillnet selectivity analyses as a relationship between swimming speed and fish length [15, 17, 18]. Rudstam et al. [17] considered encounter probability to be proportional to the length-specific swimming speed of bloater *Coregonus hoyi* in an analysis of gillnet selectivity for a closely related species, cisco *Coregonus artedii*. Incorporation of encounter probability is suggested to improve estimates of gillnet selectivity, thereby resulting in more accurate estimates of population structure [12, 15, 17]. However, it is unclear if estimating encounter probability as a length-related function leads to more accurate estimates of the length distribution of a given fish population.

Due to the widespread use of gillnet selectivity corrections and the numerous knowledge gaps surrounding their application, we evaluated the utility of correcting gillnet-derived data for estimating common population metrics. Specifically, we assessed if established gillnet selectivity corrections lead to more accurate estimates of total annual mortality and von Bertalanffy parameters for four common sport fish species. Additionally, we evaluated the value of incorporating generalized encounter probabilities for improving estimates derived from "corrected" length structure data.

## Materials and methods

### Population model

In an effort to understand how adjusting for gill net selectivity influenced population parameter estimates, we simulated populations of channel catfish *Ictalurus punctatus*, lake trout *Salvelinus namaycush*, walleye *Sander vitreus*, and white crappie *Pomoxis annularis* using known parameters (Table 1 [21, 22]). Each hypothetical population was then sampled using selectivity functions estimated from observed catch data (Table 2 [13, 14]). Common population parameters (i.e., mortality, von Bertalanffy parameters) were estimated from the resulting sample data. Estimates from uncorrected data and data corrected for gillnet selectivity were compared to evaluate if corrected gillnet data resulted in more accurate population parameter estimates.

Each population was simulated based on age-structured Leslie-matrix population models parameterized with literature-derived vital rates (Table 1 [21, 22]). The stable age distribution of a projection matrix was multiplied by 100,000 to generate initial age-specific abundances for each simulation [22]. Simulations assumed stochastic recruitment and survival. Recruitment was modeled as a random draw from a Poisson distribution of species-specific fertility (Table 1). Fertility was modeled as

$$F_t \sim Poisson\left(f_l \cdot m_t \cdot p_f \cdot S_0\right)$$

where $f_l$ is the mean fecundity at a given length, $m_t$ is the probability of maturity at age $t$, $p_f$ is the proportion of female offspring (0.5), and $S_0$ is the survival rate of age-0 fish of each species (Table 1 [23]). Survival rates of age-0 fish were unavailable for channel catfish, lake trout, and white crappie. Therefore, daily survival rates of larval fish were used to approximate survival of age-0 fish. In all instances, the daily survival rates of larval fish led to unrealistically low estimates of annual survival (daily survival[365 days]). Lorenzen and Camp [24] suggested that density-dependent mortality is strongest on fish that have not recruited to a population and are smaller that 10% of a population's asymptotic length ($L_\infty$). As such, age-0 survival was estimated as the daily survival rate expanded by the number of days for a species to reach 10% of a population's asymptotic length. Adult survival was modeled as a random draw from a binomial distribution of species-specific survival rate where abundance was the number of trials in the binomial distribution and $p$ was the probability of survival (Table 1). Each population was simulated over 15 years to generate realistic age structures. At 15 years, length-at-age was assigned to each fish in a population based on species-specific von Bertalanffy growth functions (Table 1)

$$L = L_\infty \left(1 - e^{-K(a-t_0)}\right)e^{\varepsilon l}$$

where $L_\infty$ is the average maximum length of fish in a population, $K$ is the growth coefficient, $t$ is age, $t_0$ is the theoretical age when length equals 0, and $\varepsilon_l$ is the random error assuming a normal distribution with a mean of zero and variance of 2 [25, 26].

### Sampling

One hundred and fifty hypothetical fish were sampled from each replicate population at year fifteen of the simulation under three different gillnet-catch scenarios. Although sample size is an important consideration when estimating population parameters, our goal was to compare estimates from adjusted and unadjusted age (or length) data, rather than comparing the accuracy of a specific population parameter estimate. Each hypothetical fish population was sampled assuming 1) no size-selectivity (simple random sample), 2) length-specific retention probabilities, and 3) length-specific retention and encounter probabilities. In addition, we

**Table 1. Parameters used to simulate hypothetical populations of channel catfish, lake trout, walleye, and white crappie.** The published sources of each parameter are included.

| Parameter | Value | Source |
|---|---|---|
| | **Channel catfish** | |
| Maximum age | 20 | [39] |
| Von Bertalanffy coefficients | $L_\infty = 596.00$; $k = 0.27$; $t_0 = 0.15$ | [40] |
| Probability of maturity | 0.10 (age 2), 0.50 (age 3), 0.75 (age 4+) | [41] |
| Fecundity | $2.78(\log_{10} TL) - 3.18$ | [42] |
| Age-0 survival | 0.95 day$^{-1}$ | [43] |
| Adult survival | 0.59 (age 1+) | [44] |
| | **Lake trout** | |
| Maximum age | 20 | [45] |
| Von Bertalanffy coefficients | $L_\infty = 781.00$; $k = 0.18$; $t_0 = 0.16$ | [45] |
| Probability of maturity | $\dfrac{e^{-21+0.04(TL)}}{1+e^{-21+0.04(TL)}}$ | [45] |
| Fecundity | $3.63(\log TL) - 15.08$ | [45] |
| Age-0 survival | 0.99 day$^{-1}$ | [46] |
| Adult survival | 0.45 (age 1), 0.60 (age 2+) | [47] |
| | **Walleye** | |
| Maximum age | 10 | [48] |
| Von Bertalanffy coefficients | $L_\infty = 610.00$; $k = 0.30$; $t_0 = 0.148$ | [48] |
| Probability of maturity | 0.12 (age 4), 0.55 (age 5), 0.71 (age 6), 1.00 (age 7+) | [49] |
| Fecundity | $8{,}900(TL) - 101{,}100$ | [49] |
| Age-0 survival | $4.8 \times 10^{-4}$ | [50] |
| Adult survival | 0.12 (age 1–3), 0.53 (age 4+) | [51] |
| | **White crappie** | |
| Maximum age | 10 | [52] |
| Von Bertalanffy coefficients | $L_\infty = 230.00$; $k = 0.92$; $t_0 = 0.37$ | [52] |
| Probability of maturity | $1 - \dfrac{e^{8.9968-0.0506(TL)}}{1+e^{8.9968-0.0506(TL)}}$ | [52] |
| Fecundity | $211.65(TL) + 4{,}119$ | [53] |
| Age-0 survival | 0.88 day$^{-1}$ | [54] |
| Adult survival | 0.68 (age 1+) | [52] |

assumed equal fishing intensity among mesh sizes to simplify analysis [12]. Length-specific retention probabilities were based on published species-specific retention selectivity curves (Table 2 [13, 14]). For the third scenario, the sample (catch) was the product of the retention probability and the encounter probability.

Encounter probabilities are not available for most species; however, encounter probabilities can be approximated as a length-specific swimming speed of a given species [15, 17, 18]. Encounter probabilities are related to fish length as

$$P(E_l) = X \cdot l^z$$

where $X$ is a constant and $z$ is the exponent expression for sustained swimming speed. Unfortunately, length-swimming speed relationships are not available for most fish species; therefore, we used a generalized length-swimming speed relationship to account for encounter probability. Previous research suggested that sustained swimming speeds for most species are proportional to body length raised to a power between 0.5 and 0.8 [17]. Because the swimming speed of fish in the current study was unknown, we used a general power function of 0.5 to

**Table 2. Gillnet selectivity functions for channel catfish, lake trout, walleye, and white crappie.** Selectivity functions and associated parameters for channel catfish, walleye, and white crappie were collected from Shoup and Ryswyk [14]. Selectivity functions and associated parameters for lake trout were collected from Hansen et al. [13]. The selectivity curves equations relate the mesh size $j$ ($m_j$) to the number of fish of length $l$ capture in that mesh size. The remaining parameters are constants and are defined.

| Selectivity function | Parameters |
|---|---|
| **Channel catfish** | |
| $e\left(-\frac{\left(l-k_1\times m_j\right)^2}{2k_2^2\times m_j^2}\right) + c\,e\left(-\frac{\left(l-k_3\times m_j\right)^2}{2k_4^2\times m_j^2}\right)$ | $k_1 = 10.04;$ $k_2 = 1.28;$ $k_3 = 14.75;$ $k_4 = 6.05;$ $c = 0.28$ |
| **Lake trout** | |
| $\frac{1}{\left(\sigma_0+\sigma_1 m_j\right)(2\pi)^{\frac{1}{2}}} e^{\frac{\left(l-\left(\mu_0+\mu_1 m_j\right)\right)^2}{2\left(\sigma_0+\sigma_1 m_j\right)^2}} \left(1 - \frac{1}{2q_0\left(\sigma_0+\sigma_1 m_j\right)^{3/2}}\left(\frac{\left(l-\left(\mu_0+\mu_1 m_j\right)\right)}{\left(\sigma_0+\sigma_1 m_j\right)} - \frac{\left(l-\left(\mu_0+\mu_1 m_j\right)\right)^3}{3\left(\sigma_0+\sigma_1 m_j\right)^3}\right)\right)$ | $\mu_0 = 428.539;$ $\mu_1 = 1.623;$ $\sigma_0 = 141.419;$ $\sigma_1 = -0.494;$ $q_0 = 0.045$ |
| **Walleye** | |
| $e\left(-\frac{\left(l-k_1\times m_j\right)^2}{2k_2^2\times m_j^2}\right) + c\,e\left(-\frac{\left(l-k_3\times m_j\right)^2}{2k_4^2\times m_j^2}\right)$ | $k_1 = 10.97;$ $k_2 = 1.00;$ $k_3 = 13.62;$ $k_4 = 3.32;$ $c = 0.55$ |
| **White crappie** | |
| $e\left(-\frac{\left(l-k_1\times m_j\right)^2}{2k_2^2\times m_j^2}\right) + c\,e\left(-\frac{\left(l-k_3\times m_j\right)^2}{2k_4^2\times m_j^2}\right)$ | $k_1 = 6.48;$ $k_2 = 0.61;$ $k_3 = 9.16;$ $k_4 = 3.27;$ $c = 0.18$ |

represent $z$ [18]. The term $X$ was unknown but was scaled by assuming that the largest fish in each hypothetical population had the highest probability of encountering gill nets [15, 17, 18].

Each random sample generated under scenarios 2 and 3 was adjusted to account for gillnet selectivity. Relative retention selectivity was estimated as

$$S_l = \sum_j \left(\frac{s_j(l)}{max_l}\right)$$

where $s_j(l)$ is the retention probability of length class $l$ in mesh size $j$, and $max_l$ is the maximum retention probability observed among all length classes [13–15]. For scenario 3, relative retention selectivity was adjusted for encounter probability as

$$S_l = P(E_l)\sum_j \left(\frac{s_j(l)}{\max_l}\right)$$

where $P(E_l)$ is the encounter probability of length class $l$ [15, 17]. For scenarios 2 and 3, the overall relative selectivity was used to adjust the observed count by dividing the observed number of fish in each 1-cm length bin by the overall relative selectivity of that length bin [12]. Observed counts were only adjusted for length classes originally used to estimate retention selectivity curves [5, 12, 27].

## Simulations

We conducted 1,000 replicates for each species and selectivity scenario. The instantaneous mortality rate ($z$) of each replicate population and scenario was estimated with a catch curve of the most abundant age class to the oldest observed age class [28–30]. Instantaneous mortality rates were converted to annual mortality rates as $A = 1 - e^{-z}$ [28]. In addition, von Bertalanffy models were fitted to the age and length data of each simulated population at each iteration. Average estimates of $A$, $K$, and $L_\infty$ were compared to the true (simulated) parameters of the population to understand the utility of correcting for gillnet selectivity using the indirect method. We assumed estimates of $K$ and $L_\infty$ would highlight any important patterns associated with gillnet selectivity adjustments and excluded estimates of $t_o$ for brevity. All analysis was conducted using R statistical software [31]. The "popbio" package was used to estimate the stable age distribution of each population [32].

## Results

Relative retention probabilities varied by species (Fig 1). Channel catfish, walleye, and white crappie were bimodal, whereas lake trout had a unimodal retention probability. Relative retention was high for most length classes regardless of species. For instance, 50% to 75% of the length classes considered had a retention probability greater than 60%. Unsurprisingly, incorporation of encounter probability reduced the overall selectivity for all species (Fig 1). Maximum reductions in selectivity varied from 14% to 30% and generally occurred at smaller length classes. At larger length classes, reductions in selectivity were less apparent. For instance, overall selectivity declined by 3% to 22% at peak relative retentions.

Accounting for relative retention probabilities rarely improved estimates of total annual mortality, $K$, or $L_\infty$ (Figs 2–4). For instance, estimates of annual mortality of lake trout based on uncorrected data were overestimated by 0.2%, whereas adjusted age data resulted in an overestimate of $A$ by 13.7%. Similar increases in inaccuracy of annual mortality estimates following adjustment of age data were apparent for walleye (4% increase) and white crappie (0.4% increase). Conversely, channel catfish exhibited a slight improvement in the accuracy of annual mortality estimates following adjustment of age data (21.7% to 22.2%). Adjusting for relative retention probability resulted in very small changes in estimates of $K$ and $L_\infty$ for all species (Figs 3 and 4). For example, changes in $K$ varied from 0.0000038 to 0.0209884 for all species following adjustment for relative retention probability. Similarly, the largest change in mean estimates of $L_\infty$ was for lake trout (0.96), whereas the smallest change in $L_\infty$ was for walleye (0.0019). These small changes in estimates of $K$ and $L_\infty$ never resulted in meaningful improvements in the accuracy of a given estimate. For instance, an increase of 0.96 for $L_\infty$ would result in an increase in the estimated length of an age-2 lake trout of 0.31 mm, assuming $k = 0.18$ and $t_0 = 0.16$.

Adjusting for encounter probability resulted in similar patterns to those of relative retention probability (Figs 2–4). For instance, inaccuracies in estimates of annual mortality were magnified for lake trout (14.4%) and walleye (3.0%) following adjustment for encounter probability (Fig 2). Interestingly, adjusting for encounter probability resulted in improved estimates of annual mortality for channel catfish and white crappie, although the improvements were generally small (0.6–0.8%; Fig 2). Estimates of $K$ and $L_\infty$ only exhibited small changes after adjustment for encounter probability (Figs 3 and 4). For instance, mean estimates of $L_\infty$ only changed by 0.001 to 1.002 among all species after accounting for encounter probability. Similarly, mean estimates of $K$ varied from 0.0000032 to 0.0211174 after accounting for encounter probability. These small changes in the mean estimates of $K$ and $L_\infty$ never improved the accuracy of a given estimate relative to the true value.

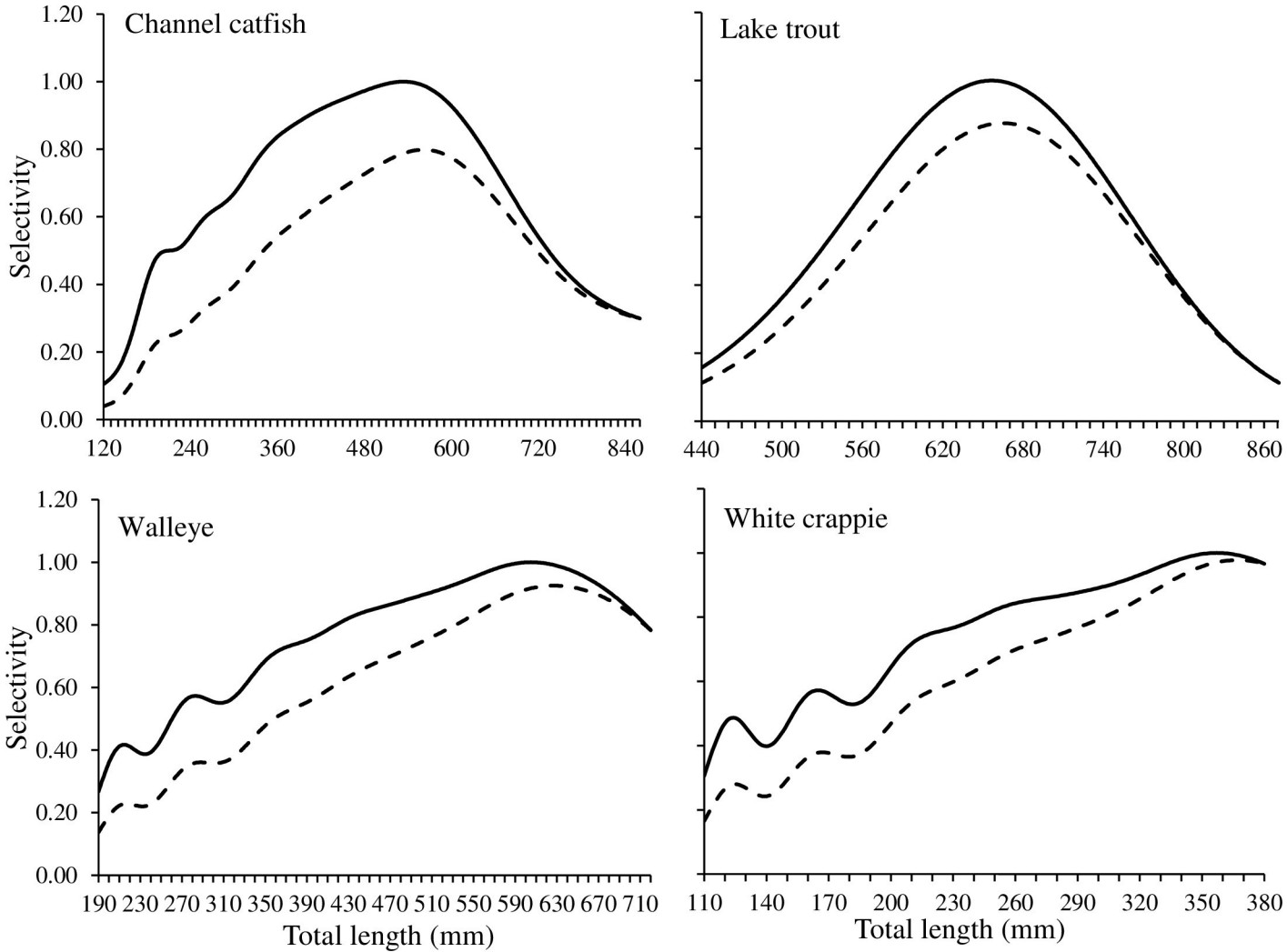

**Fig 1. Selectivity curves for channel catfish, lake trout, walleye, and white crappie.** The solid lines represent relative retention selectivity and the dotted lines represent retention and encounter selectivity. Note that x-axes have different scales.

Accounting for encounter probability generally improved estimates compared to adjusting age and length data for relative retention probability alone (Figs 2–4). Mean annual mortality estimates from adjusted encounter probability data resulted in improved estimates for lake trout, walleye, and white crappie (Fig 2). However, the improvements were generally small (0.7% - 2.2%). For example, adjusting encounter probability data for channel catfish improved the mean annual mortality estimate by about 2.2% but still underestimated annual mortality by about 16.0%. The value of adjusting for encounter probability compared to relative retention probability was much less apparent for estimates of $K$ and $L_\infty$. Accounting for encounter probability resulted in very small changes in average estimates of $K$ (0.000–0.005) and $L_\infty$ (0.033–3.278) for all species.

Assuming no selectivity (random sample) often outperformed estimates from adjusted age and length data for $K$ and $L_\infty$, however the improvements were generally negligeable ($< 0.5$; Figs 3 and 4). Similar patterns were not apparent for estimates of mean annual mortality (Fig 2). For instance, data from the no selectivity simulations resulted in less accurate estimates of

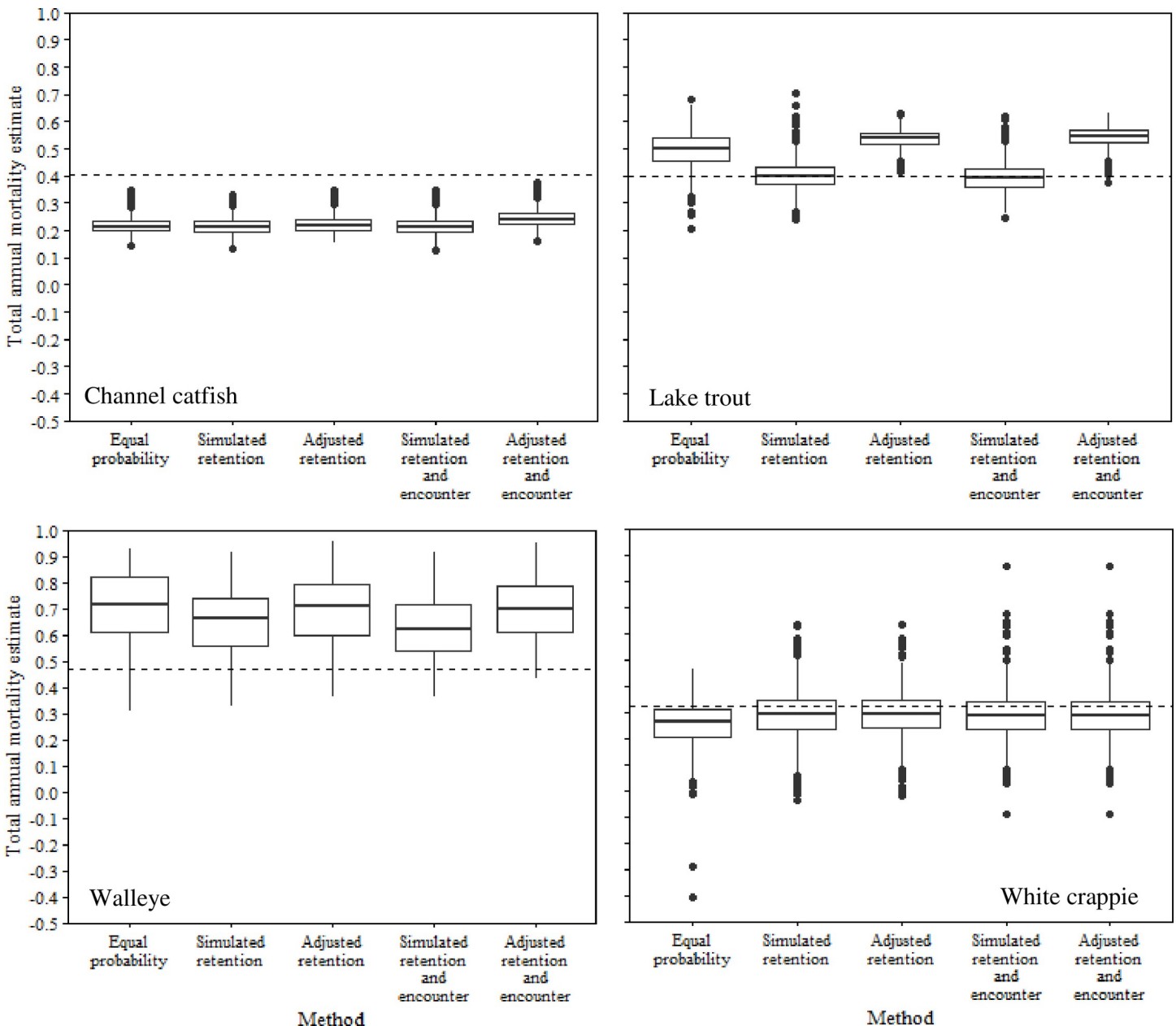

**Fig 2. Box plots of total annual mortality estimates from simulated populations of channel catfish, lake trout, walleye, and white crappie.** Annual mortality was estimated using age data derived from simulations assuming 1) equal probability of capture, 2) capture based on retention probability, and 3) capture based on relative retention and encounter probabilities. Adjusted estimates represent age data that were corrected for relative retention probability or relative retention and encounter probabilities. The dashed lines represent the true annual mortality rate for each simulated population.

annual mortality for channel catfish, walleye, and white crappie compared to estimates from adjusted data.

## Discussion

Various authors have suggested that accounting for selectivity will lead to more accurate estimates of population parameters [7, 14, 15, 33]. Smith et al. [7] suggested that data collected

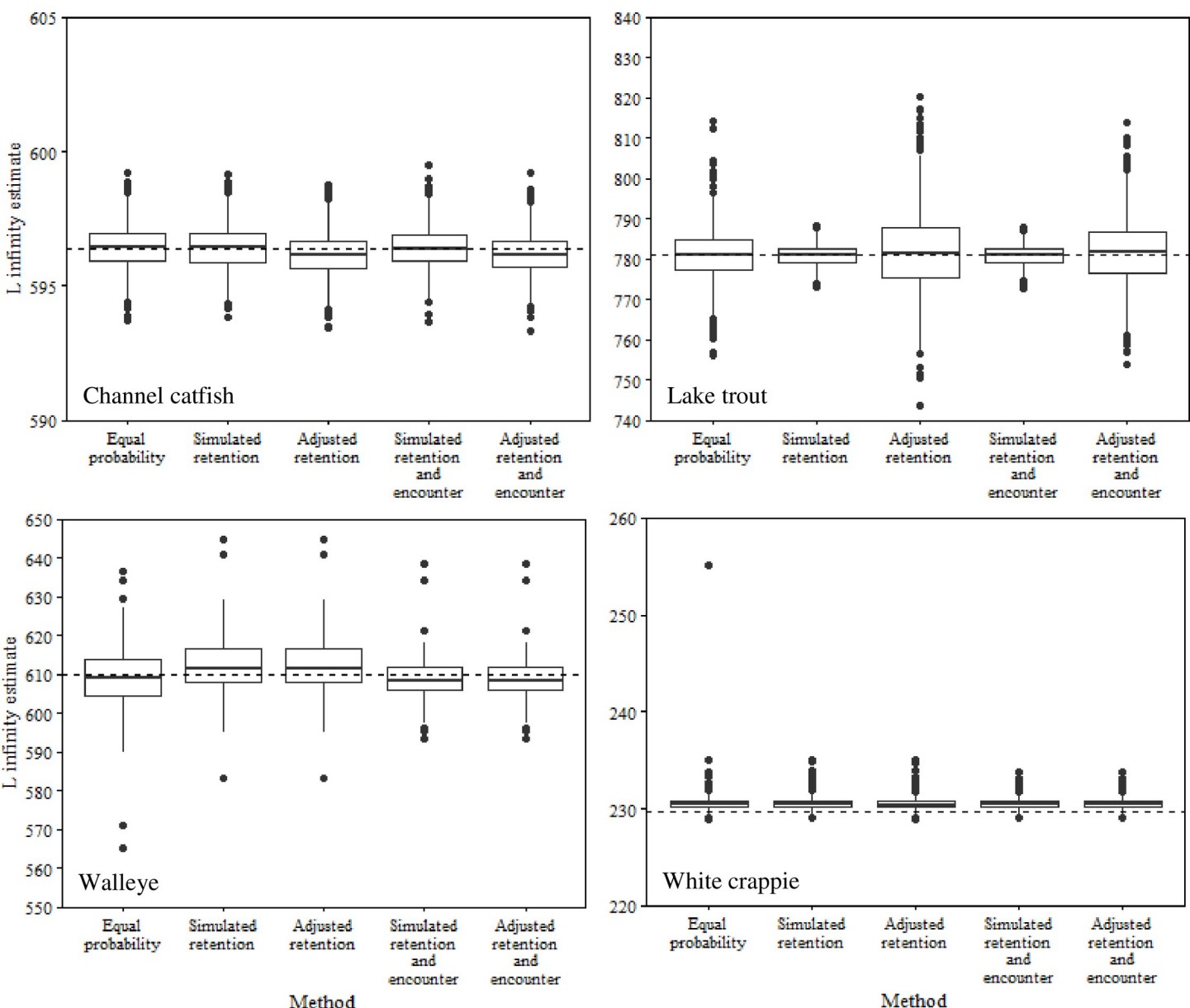

**Fig 3. Box plots of *L* infinity estimates from simulated populations of channel catfish, lake trout, walleye, and white crappie.** *L* infinity was estimated using length-at-age data derived from simulations assuming 1) equal probability of capture, 2) capture based on retention probability, and 3) capture based on relative retention and encounter probabilities. Adjusted estimates represent length-at-age data that were corrected for relative retention probability or relative retention and encounter probabilities. The dashed lines represent the true *L* infinity value for each simulated population.

using North American standard gill nets should be corrected for contact selectivity (i.e., retention probability) to provide more reliable information on size structure and related indices. However, our results indicate that accounting for retention and encounter probabilities does not substantially improve estimates of mortality using catch curves or growth parameters using von Bertalanffy growth modes. Adjustments for retention and encounter probability generally resulted in less accurate estimates of $A$, $K$, or $L_\infty$ compared to estimates from unadjusted data in the current study. In certain instances, adjusting for relative retention and encounter probabilities improved estimated values (i.e., channel catfish, white crappie),

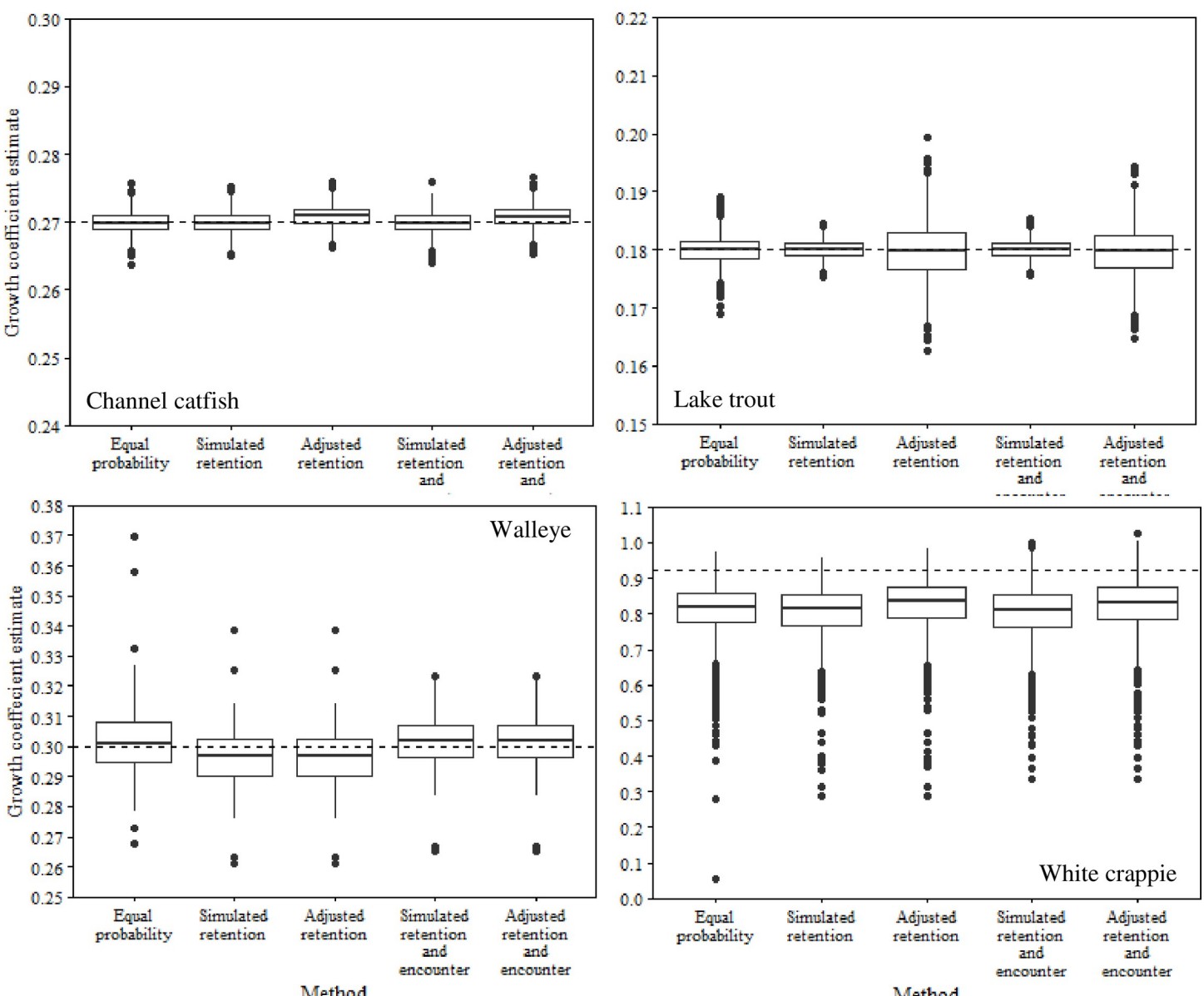

**Fig 4. plots of von Bertalanffy growth coefficients from simulated populations of channel catfish, lake trout, walleye, and white crappie. Box** Growth coefficients were estimated using length-at-age data derived from simulations assuming 1) equal probability of capture, 2) capture based on retention probability, and 3) capture based on relative retention and encounter probabilities. Adjusted estimates represent length-at-age data that were corrected for relative retention probability or relative retention and encounter probabilities. The dashed lines represent the true growth coefficient for each simulated population.

however the improvements were always minor and rarely resulted in meaningful improvements in the accuracy of estimated parameters. For instance, mean annual mortality estimates improved by 0.3–0.6% for white crappie after age data were adjusted for retention and encounter probability. As such, accounting for gillnet selectivity using the indirect method may not be necessary and may lead to less accurate estimates of the population parameters considered in this study.

One of the practical limitations of indirect estimation studies is that most selection curves only estimate the relative retention probability and ignore other processes that likely influence

gillnet selectivity. Millar and Fryer [6] suggested capture in gill nets is the result of size-dependent processes associated with a fish's availability, encounter probability, and retention probability. Regardless of the exact description of the capture process, nearly all of the factors contributing to capture in gillnets are not known and cannot be effectively quantified without additional information [6, 11, 12]. Thus, indirect methods of estimating gillnet selectivity typically only account for the relative probability that a fish of a given length is retained in a gill net given that it has contacted the net. Because the complicated process of capture in gill nets is reduced to a relative retention probability, adjusted count data likely do not approximate the true structure of a fish population. For instance, Grant et al. [19] found that individual walleye exhibited various avoidance, escape, and contact behaviors during gillnet sampling in Moody Lake, MN. The authors concluded that an inability to account for these various behaviors in previous gillnet retention probability estimates resulted in an underestimate of peak retention by approximately 40%. The inability to accurately account for the numerous processes governing gillnet selectivity suggests that adjustments based on relative retention alone may not result in accurate estimates of population's true structure.

Accounting for encounter probability has been suggested to improve estimates of gillnet selectivity [15, 17, 18]. Rudstam et al. [17] noted that accounting for encounter probability using a length-related power function increased the proportional abundance of small fish relative to large fish, thereby better approximating the true structure of a given fish population. As such, we attempted to account for encounter probability by relating fish length to a generalized swimming speed ($X \cdot l^{0.5}$). Using this method of correcting for encounter probability did improve estimates relative to retention corrections alone, but the improvements were always minor. For example, adjusting for encounter probability resulted in an improvement of 1.6% for estimates of mean annual mortality for walleye in the current study. Using species-specific power functions would likely improve the accuracy of adjustments based on encounter probabilities. However, power functions for length-specific swimming speeds are difficult to measure and have only been quantified for a small number of species [17, 34, 35]. Furthermore, equating encounter probability to swimming speed alone assumes that other length-related behaviors (e.g., avoidance, availability) are constant among length classes [12, 17, 18]. Such assumptions may be valid in certain situations (e.g., pelagic, mobile species), but in many species gillnet-related behavior is poorly understood and adjusting gillnet-derived data under such assumptions likely leads to inaccurate population data.

Although our results indicate that adjusting length and(or) age data based on the indirect method did not lead to substantial improvements in the accuracy of the population parameters estimated in this study, that does not preclude the value of correcting for selectivity in many other applications. For instance, gillnet selectivity is an essential parameter for many types of catch-at-age models [36, 37]. Selectivity is also commonly used to understand length-related vulnerability in gillnet fisheries [6]. Carlson and Cortés [38] used relative retention probabilities to suggest the mesh sizes needed to protect juvenile and adult sharks from capture in muti-species gillnet fisheries in the Gulf of Mexico. In such instances, incorporating relative retention probabilities is a vital component of analysis. However, when estimating the metrics used in the current study, incorporating gillnet selectivity likely leads to less accurate estimates at worst and negligible improvements at best. Furthermore, estimation of gillnet selectivity requires additional data collection and analysis [12]. Thus, gillnet selectivity should likely only be estimated when specific questions are being addressed (e.g., catch-at-age models, length-specific vulnerability) or if the processes influencing size-dependent availability and avoidance are known.

## Acknowledgments

This work was supported by New Mexico State University and the Idaho Department of Fish and Game. We are especially grateful to D. Shoup for providing gillnet selectivity data. We thank D.K. Meena and two anonymous reviewers for their helpful comments on previous versions of this manuscript.

## Author Contributions

**Conceptualization:** Zachary Klein, Joshua McCormick.

**Data curation:** Zachary Klein, Joshua McCormick.

**Formal analysis:** Zachary Klein, Joshua McCormick.

**Investigation:** Zachary Klein.

**Methodology:** Zachary Klein, Joshua McCormick.

**Writing – original draft:** Zachary Klein.

**Writing – review & editing:** Zachary Klein, Joshua McCormick.

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
