## [Decision Letter · Decision Letter 0]

20 Mar 2023

PONE-D-23-04666Evaluation of the influence of correcting for gillnet selectivity on the estimation of population parametersPLOS ONE

Dear Dr. Klein,

Thank you for submitting your manuscript to PLOS ONE. After careful consideration, we feel that it has merit but does not fully meet PLOS ONE’s publication criteria as it currently stands. Therefore, we invite you to submit a revised version of the manuscript that addresses the points raised during the review process.

We look forward to receiving your revised manuscript.

Kind regards,

Dharmendra Kumar Meena

Academic Editor

PLOS ONE

Journal Requirements:

4. Please ensure that you include a title page within your main document. You should list all authors and all affiliations as per our author instructions and clearly indicate the corresponding author.

Additional Editor Comments:

The article can not be accepted in its present form . So author are suggested to resubmit it as major revisions.

Reviewers' comments:

Reviewer's Responses to Questions

**Comments to the Author**

1. Is the manuscript technically sound, and do the data support the conclusions?

Reviewer #1: Yes

Reviewer #2: Partly

2. Has the statistical analysis been performed appropriately and rigorously? 

Reviewer #1: Yes

Reviewer #2: Yes

3. Have the authors made all data underlying the findings in their manuscript fully available?

Reviewer #1: No

Reviewer #2: No

4. Is the manuscript presented in an intelligible fashion and written in standard English?

Reviewer #1: Yes

Reviewer #2: Yes

5. Review Comments to the Author

Reviewer #1: Numbers are showing along the margin continuosly which is not necessory and one place and along with coma(,) should be remove and should be clealyr mention of Figure and figs numbers while work is informative for the sustainable fisheries management.

Reviewer #2: Major comments

1) More detail needs to be added in the Methods and Results. Much of the text is vague and not detailed enough to be repeated. Specific examples are included in the minor comments below.

2) All Table and Figure captions need to be more detailed. The caption text needs to be stand-alone whereby a reader could read the caption and understand what you are presenting without needing to read the entire text.

3) The authors use only one group of population parameters for each species for their simulations. Fish populations are not uniform across water bodies which limits the understanding of how the analysis can influence different types of populations. It would be more valuable and informative to simulate different types of populations for each species (high size structure, average size structure, low size structure) in relation to mesh size used. One would expect more extreme biases in the population parameters when there is a mismatch of size structure and mesh size.

4) Much of the effectiveness of using gillnets (and all sampling gear) is that they are used effectively in relation to fish behavior to maximize catches, for example the season set, the habitat they are used in, the orientation of the nets (horizontal versus vertical). The authors briefly mention the issues with gillnet encounter probability and gillnet retention in the discussion (lines 298-315) but there should be some mention of this in the methods. Mainly that these issues exist and are a primary consideration of the gear but are outside the scope of this manuscript or too difficult to include in the simulation. So all simulations were conducted assuming optimal gill net sampling seasonality and habitat placement for the species in question to maximize encounter probability. Then assuming that encounter probability was strictly related to fish movement/swim speed, etc.

Minor comments

line 129-130 and throughout this paragraph: Be sure to cite table 1 where appropriate to connect this text to the actual values used.

line 153: why 150 fish in the sample? was there a power analysis or something associated with this sample size that met some criteria? Did you test for the effect of larger or smaller sample size and how this could effect your results?

Table 2: the parameters in these selectivity curves need to be defined. You provide the values in the table but the letter and symbols represent.

line 175-177: There is literature that discusses swimming speed for these species. Examples below. If this information isn't adequate for your needs you need to specify why and not say that it isn't available.

Peake et al. https://doi.org/10.1139/z00-097 for swimming speed of walleye.

Gaston thesis for crappie https://egrove.olemiss.edu/etd/118/.

Channel catfish https://journals.biologists.com/jeb/article/220/4/597/18666/Physostomous-channel-catfish-Ictalurus-punctatus

Lake trout https://doi.org/10.1577/T08-174.1
https://doi.org/10.1111/j.1095-8649.2000.tb00259.x

line 183: The mesh sizes that were simulated here should be discussed in more detail. Mesh size would have a large impact on retention. There needs to be more information here. Retention probability would also not be related to length alone but fish shape and other morphometrics. For example, head size and gill size relative to body size would make crappie retention and susceptibility to different mesh sizes very different than that of walleye. Additionally, channel catfish pectoral spine and spine locking may influence their gill net selection and retention differently relative to body size than for lake trout or walleye. This concept of retention should be expanded or at least considered in how body shape influences retention and ultimately your simulation results.

line 196: what were the age class sample size cutoffs considered here? be sure to be specific about these things. The same with von Bertalanffy model, mention the length bin sample cutoffs that were used (5 per length bin, 8, 10?). What was the requirement?

line 205: also cite all the R packages that were used.

line 215: how did mesh size play a role in this result? Its unclear at this point how mesh size was included in the simulations.

lines 224 - 227: Throughout the results, be sure to back up your statements with statistical output, values or citation of a figure. Text statements alone are not adequate.

line 229 - 231: How relevant are these small changes? It would help put this into context for the reader to explain the length at age output for these differences in L infinite or K. For example, "K increased from 0.2699 to 0.2709... These minor changes in K did not result in a difference in estimated length-at-age of channel catfish. A 4 year old catfish with a K of 0.2699 was estimated to be 7.0 inches whereas a 4 year old catfish with a K of 0.2709 was estimated to be 7.1 inches." Making that up of course but hopefully you get where I'm going.

line 208, 257, 261 as examples: be consistent with terminology. Figure, Fig. or Figs.? pick one style of formatting and stick with it.

line 269: 'minor' is vague. These are your results, be specific, provide values.

line 324: This walleye result wasn't stated in the results? Make sure everything is reported in the results.

6. PLOS authors have the option to publish the peer review history of their article (what does this mean?). If published, this will include your full peer review and any attached files.

Reviewer #1: **Yes: **Dr. Veerendra Singh

Reviewer #2: No

---

## [Author Response · Author response to Decision Letter 0]

17 May 2023

Dear Dharmendra Kumar Meena, 

We thank you and the reviewers for their thoughtful comments on our manuscript. In general, the manuscript was well received and required a few areas of clarification before it was deemed appropriate for publishing in PLOS ONE. We found that most the major and minor comments were associated with factors germane to the use of gill nets but were outside of the scope of the current manuscript. For instance, Reviewer 2 had several comments about the influence of mesh size and sample size on our results. Although mesh size and sample size are incredibly important considerations when using gillnets (or estimating population parameters from gillnet data), our study was primarily focused on understanding if adjusting for size selectivity resulted in more accurate populations parameter estimates. As such, mesh size and sample size were largely unimportant to our results and conclusions. For instance, the published selectivity curves used in our simulations were estimated for each species from multiple populations using standard gill nets (no change in mesh sizes). These selectivity curves were simply used in our analysis to “sample” hypothetical populations and “adjust” the resulting age (or length) data. If alternative size selectivity function were used, we would expect different age (or length) distributions in the sample but comparisons between adjusted and unadjusted estimates would remain the same. Similarly, sample size changes would likely influence the accuracy of population parameter estimates. However, we were not concerned with how accurate our population parameter estimates were relative to the true population. Instead, we were interested in understanding if adjusting for size selectivity resulted in more accurate population parameter estimates relative to not adjusting for size selectivity. We understand that much of this confusion may be due to a lack of clarity in our manuscript and have amended the text to alleviate future confusion. The majority of the additional comments were relatively minor and have been addressed in the text and below. We believe we have adequately addressed the concerns of both reviewers resulting in a much-improved manuscript. We again thank you and the reviewers for the thoughtful comments and look forward to hearing from you. 

Sincerely, 

Zachary Klein

Response: We will ensure out files conform to the requirements of PLOS ONE.

Response: We apologize for the confusion, but we believe we have included the necessary data needed to recapitulate our result. Specifically, the data contained in Tables 1 and 2 are the only data used in our simulations.

4. Please ensure that you include a title page within your main document. You should list all authors and all affiliations as per our author instructions and clearly indicate the corresponding author.

Response: We apologize for our error. We will submit and appropriate title page during resubmission. 

Reviewer #1: Numbers are showing along the margin continuosly which is not necessory and one place and along with coma(,) should be remove and should be clealyr mention of Figure and figs numbers while work is informative for the sustainable fisheries management.

Response: The PLOS ONE submission guidelines state “Include page numbers and line numbers in the manuscript file. Use continuous line numbers (do no restart the numbering on each page).”. Therefore, we included continuous line numbering in the manuscript. 

Reviewer #2: Major comments

1) More detail needs to be added in the Methods and Results. Much of the text is vague and not detailed enough to be repeated. Specific examples are included in the minor comments below.

Response: We apologize for the lack of detail. Based on the minor comments below, we have added text to clarify our methods and results. 

2) All Table and Figure captions need to be more detailed. The caption text needs to be stand-alone whereby a reader could read the caption and understand what you are presenting without needing to read the entire text.

Response: We have added additional detail to the table and figure captions. 

3) The authors use only one group of population parameters for each species for their simulations. Fish populations are not uniform across water bodies which limits the understanding of how the analysis can influence different types of populations. It would be more valuable and informative to simulate different types of populations for each species (high size structure, average size structure, low size structure) in relation to mesh size used. One would expect more extreme biases in the population parameters when there is a mismatch of size structure and mesh size.

Response: The goal of this research was to assess if adjusting for size selectivity resulted in more accurate estimates of mortality and Von Bertalanffy parameters, rather than evaluating how a mismatch between mesh size and size structure influence resulting populations metric estimates. To do this, we simulated four fish populations and “sampled” fish using published selectivity functions. The selectivity functions used were based on length-mesh size relationships estimated from a number of existing fish populations. For instance, the selectivity function for White Crappie (Table 2 in the manuscript) was estimated from five crappie populations in Oklahoma, USA (Shoup and Ryswyk 2016). As such, the selectivity functions used in the current study incorporated variability in the size structure of each of the four fish populations. Because the selectivity functions used in the current study were estimated from numerous populations, length-mesh relationships should be robust to any potential biases in mismatch between size structure and mesh. Furthermore, our research was not focused on evaluating how well gill nets represent the size structure of a fish populations. Instead, we were interested in understanding how adjusting for size selectivity would influence populations parameter estimates compared to not adjusting for size selectivity. 

4) Much of the effectiveness of using gillnets (and all sampling gear) is that they are used effectively in relation to fish behavior to maximize catches, for example the season set, the habitat they are used in, the orientation of the nets (horizontal versus vertical). The authors briefly mention the issues with gillnet encounter probability and gillnet retention in the discussion (lines 298-315) but there should be some mention of this in the methods. Mainly that these issues exist and are a primary consideration of the gear but are outside the scope of this manuscript or too difficult to include in the simulation. So all simulations were conducted assuming optimal gill net sampling seasonality and habitat placement for the species in question to maximize encounter probability. Then assuming that encounter probability was strictly related to fish movement/swim speed, etc.

Response: Thank you for the comment. We agree numerous factors should be considered when setting gillnets. However, we disagree that these factors are related to retention probabilities. Retention probabilities assume that a fish has already encountered the gear and simple quantifies the probability that the fish will be retained by the gear. However, we do agree that retention probability is likely influenced by fish behavior, which can be indirectly influenced by factors such as season or habitat. As such, we have added text clarifying that encounter probability can be influenced by numerous factors such as season and habitat.

Minor comments

line 129-130 and throughout this paragraph: Be sure to cite table 1 where appropriate to connect this text to the actual values used.

Response: We have made the suggested changes

line 153: why 150 fish in the sample? was there a power analysis or something associated with this sample size that met some criteria? Did you test for the effect of larger or smaller sample size and how this could effect your results?

Response: We selected 150 fish because it reflects a reasonable number of fish that would be sampled in routine fish monitoring. That being said, the number of fish sampled is immaterial for the comparisons in the current study. The goal of the current research was to assess the value of adjusting for size selectivity in gill nets. Therefore, we compared population parameter estimates from “adjusted” to “unadjusted” age (or length) structure data. Increasing the sample size would reduce the variance of an individual estimate, but would not change comparisons between estimates. 

Table 2: the parameters in these selectivity curves need to be defined. You provide the values in the table but the letter and symbols represent.

Response: We have defined each of the parameters in the selectivity curves.

line 175-177: There is literature that discusses swimming speed for these species. Examples below. If this information isn't adequate for your needs you need to specify why and not say that it isn't available.

Peake et al. https://doi.org/10.1139/z00-097 for swimming speed of walleye.

Gaston thesis for crappie https://egrove.olemiss.edu/etd/118/.

Channel catfish https://journals.biologists.com/jeb/article/220/4/597/18666/Physostomous-channel-catfish-Ictalurus-punctatus

Lake trout https://doi.org/10.1577/T08-174.1
https://doi.org/10.1111/j.1095-8649.2000.tb00259.x

Response: Thank you for the comment. We agree that there are numerous examples of swimming speed in fish in the literature. However, much of this work lacks the swimming speed – length relationship that is critical for estimating a generalized encounter probability. We have clarified this issue in the text.

line 183: The mesh sizes that were simulated here should be discussed in more detail. Mesh size would have a large impact on retention. There needs to be more information here. Retention probability would also not be related to length alone but fish shape and other morphometrics. For example, head size and gill size relative to body size would make crappie retention and susceptibility to different mesh sizes very different than that of walleye. Additionally, channel catfish pectoral spine and spine locking may influence their gill net selection and retention differently relative to body size than for lake trout or walleye. This concept of retention should be expanded or at least considered in how body shape influences retention and ultimately your simulation results.

Response: As stated above, the selectivity functions used in the current study were estimated from populations of each species. Thus, the selectivity functions accounted for fish shape (and other morphometrics) on the retention probability of a given species. For instance, the bimodal selectivity functions used by (Shoup and Ryswyk 2016) were developed to specifically account for fish caught via different modes (e.g., “gilling”, tangling). 

line 196: what were the age class sample size cutoffs considered here? be sure to be specific about these things. The same with von Bertalanffy model, mention the length bin sample cutoffs that were used (5 per length bin, 8, 10?). What was the requirement?

Response: Our simulations represented a “real-world” sampling event, whereby gill nets sampled 150 fish based on published retention probabilities. As such, fish from individual age classes were sampled randomly and did not have a “set” sample size. We agree that sample size can influence the overall accuracy of population parameter estimates. However, assessing sample size needs for accurate estimates of population parameters was not the goal of the research. Rather, we were interested in understanding if adjusting for size selectivity resulted in more accurate populations parameter estimates relative to not adjusting for size selectivity. 

line 205: also cite all the R packages that were used.

Response: We have made the suggested revision. 

line 215: how did mesh size play a role in this result? Its unclear at this point how mesh size was included in the simulations.

Response: Mesh size did not play a role in simulations. We used published selectivity functions that were developed for a given species using particular mesh sizes. Therefore, these selectivity functions are only applicable for a particular species and gill nets configuration (i.e., mesh sizes). It is well understood that selectivity functions should not be applied to species sampled with a different size of mesh because the length-mesh relationships do not apply to mesh sizes outside of those used to estimate the original selectivity function (Millar 2000).

Millar RB. Untangling the confusion surrounding the estimation of gillnet selectivity. Can J Fish Aquat Sci. 2000;57:507-11.

 lines 224 - 227: Throughout the results, be sure to back up your statements with statistical output, values or citation of a figure. Text statements alone are not adequate.

Response: We have added more statistics to our results. 

line 229 - 231: How relevant are these small changes? It would help put this into context for the reader to explain the length at age output for these differences in L infinite or K. For example, "K increased from 0.2699 to 0.2709... These minor changes in K did not result in a difference in estimated length-at-age of channel catfish. A 4 year old catfish with a K of 0.2699 was estimated to be 7.0 inches whereas a 4 year old catfish with a K of 0.2709 was estimated to be 7.1 inches." Making that up of course but hopefully you get where I'm going.

Response: Thank you for the comment. We have added text clarifying the relevancy of the small changes in our population parameter estimates. 

line 208, 257, 261 as examples: be consistent with terminology. Figure, Fig. or Figs.? pick one style of formatting and stick with it.

Response: We apologize for the error and have ensured that terminology is consistent throughout the text. 

line 269: 'minor' is vague. These are your results, be specific, provide values. 

Response: We have made the suggested changes.

line 324: This walleye result wasn't stated in the results? Make sure everything is reported in the results.

Response: We have made the suggested changes. 

---

## [Decision Letter · Decision Letter 1]

7 Jun 2023

Evaluation of the influence of correcting for gillnet selectivity on the estimation of population parameters

PONE-D-23-04666R1

Dear Dr. Klein,

We’re pleased to inform you that your manuscript has been judged scientifically suitable for publication and will be formally accepted for publication once it meets all outstanding technical requirements.

Kind regards,

Dharmendra Kumar Meena

Academic Editor

PLOS ONE

Additional Editor Comments (optional):

The article can be accepted provided with completion of the minor revisions as suggested by the reviewer.

Reviewers' comments:

Reviewer's Responses to Questions

**Comments to the Author**

1. If the authors have adequately addressed your comments raised in a previous round of review and you feel that this manuscript is now acceptable for publication, you may indicate that here to bypass the “Comments to the Author” section, enter your conflict of interest statement in the “Confidential to Editor” section, and submit your "Accept" recommendation.

Reviewer #1: All comments have been addressed

Reviewer #3: (No Response)

2. Is the manuscript technically sound, and do the data support the conclusions?

Reviewer #1: Yes

Reviewer #3: Yes

3. Has the statistical analysis been performed appropriately and rigorously? 

Reviewer #1: Yes

Reviewer #3: Yes

4. Have the authors made all data underlying the findings in their manuscript fully available?

Reviewer #1: No

Reviewer #3: (No Response)

5. Is the manuscript presented in an intelligible fashion and written in standard English?

Reviewer #1: Yes

Reviewer #3: Yes

6. Review Comments to the Author

Reviewer #1: As we know that Gill Net is passive gear and it's have so many types or forms of itself like as triple wall or trammel net or bottom set gill net so which kind of gill net you have used for the study of growth parameters or mortality please mention it and bell shape curve is assumed for the lake trout only and for others species it's show irregularities and when we use age data for the estimation of Total Mortalities then it only applicable for tropical species.

Reviewer #3: 1. References in list and in text should be as per journal format. Ensure the same.

2. In the methods section, authors need to mention their type of sampling, whether random, purposive???

3. The authors need to justify how their sample size (150 in the present case) is enough for the present study and to arrive at definite conclusions.

4. Conclusion section to be added based on the findings.

7. PLOS authors have the option to publish the peer review history of their article (what does this mean?). If published, this will include your full peer review and any attached files.

Reviewer #1: **Yes: **Dr. Veerendra Singh

Reviewer #3: No

---

## [Editor Report · Acceptance letter]

13 Jun 2023

PONE-D-23-04666R1 

Evaluation of the influence of correcting for gillnet selectivity on the estimation of population parameters 

Dear Dr. Klein:

I'm pleased to inform you that your manuscript has been deemed suitable for publication in PLOS ONE. Congratulations! Your manuscript is now with our production department. 

Kind regards, 

on behalf of

Dr. Dharmendra Kumar Meena 

Academic Editor

PLOS ONE